# Humans diverge from language models when predicting spoken language

**Thomas L. Botch**
Dartmouth College
{thomas.l.botch}@dartmouth.edu

**Emily S. Finn**
Dartmouth College
{emily.s.finn}@dartmouth.edu

## Abstract

Humans communicate through both spoken and written language, often switching between these modalities depending on their goals. The recent success of large language models (LLMs) has driven researchers to understand the extent to which these models align with human behavior and neural representations of language. While prior work has shown similarities in how humans and LLMs form predictions of written text, no work has investigated whether LLMs are representative of human predictions of spoken language. We investigated the alignment between LLMs and behavior of human participants (N=300) who predicted words within a story presented as either spoken language or written text. We found that LLM predictions were more similar to humans' predictions of written text compared to spoken language, though humans' predictions of spoken language were the most accurate. Then, by training encoding models to predict neural activity recorded with fMRI to the same auditory story, we showed that models based on human predictions of spoken language better aligned with observed brain activity during listening compared to models based on LLM predictions. These findings suggest that the structure of spoken language carries additional information relevant to human behavior and neural representations.

## 1 Introduction

Language is a flexible medium of communication, where underlying symbolic structures retain their meaning whether spoken or written (Rubin et al., 2000; Louwerse, 2011). Humans regularly switch between these modalities (Chafe & Tannen, 1987; Rubin, 1987; Hulme & Snowling, 2014) and represent these structures through a common neural code (Regev et al., 2013; Deniz et al., 2019). Recently, large language models (LLMs) have provided researchers with tools to probe the functions that underpin efficient processing and representation of language (Linzen & Baroni, 2021). A number of studies have demonstrated similarities in LLM representations of language and human neural representations of both spoken and written language (Schrimpf et al., 2021; Caucheteux & King, 2022; Caucheteux et al., 2022; Tang et al., 2022; Heilbron et al., 2022; Toneva et al., 2022). Importantly, the mechanisms by which LLMs process and predict language relate not only to human neural representations, but also to human behavior. LLM surprisal — the uncertainty in predicting an upcoming word — has been shown to predict a variety of human behaviors including reading times (Hao et al., 2020; Wilcox et al., 2020; Shain et al., 2022), event segmentation (Kumar et al., 2023; Michelmann et al., 2023), and next-word predictions of written text (Jacobs & McCarthy, 2020; Goldstein et al., 2022b). In light of these apparent similarities, it is critical to understand the extent to which LLMs recapitulate human behavior and representations of language, particularly when these models are used in downstream analyses of neural representations.

Recent interdisciplinary work has started to characterize how LLMs align with and diverge from human behavior and neural representations (Momennejad, 2023; Sucholutsky et al., 2023). Many differences between LLMs and human behavior have been reported, most often in tasks involving higher-order cognitive processes such as social reasoning (Mahowald et al., 2023; Ullman, 2023) and moral judgment (Jiang et al., 2022; Jin et al., 2022). Yet only a few studies have directly compared how these models relate to human performance within the central task used for LLM training: next-word prediction. While LLM next-word predictions often align with human predictions of commonly occurring words (Goldstein et al., 2022a), this alignment deteriorates when predicting

complex words (Jacobs & McCarthy, 2020) and when re-experiencing the same stimulus (Vaidya et al., 2023). However, no research has examined whether LLM predictions of language are equally aligned to humans' predictions of both spoken and written language.

Here, we asked human participants to make next-word predictions during a real-world story presented as either spoken or written language. We leveraged these predictions to evaluate differences in humans' *behavioral alignment* to LLMs based on the modality of the stimulus. We found that human predictions were more semantically similar to the ground-truth word than LLM predictions, regardless of the modality of presentation, and this difference was exaggerated when LLMs had high uncertainty in their predictions (i.e., high surprisal). We then used these predictions to assess *representational alignment* — specifically, whether human predictions of spoken language are more closely aligned to neural activity than LLM predictions of the same story. These behavioral differences in next-word prediction were recapitulated in their alignment to neural representations, such that human predictions of spoken language were more predictive of brain activity than words predicted by LLMs.

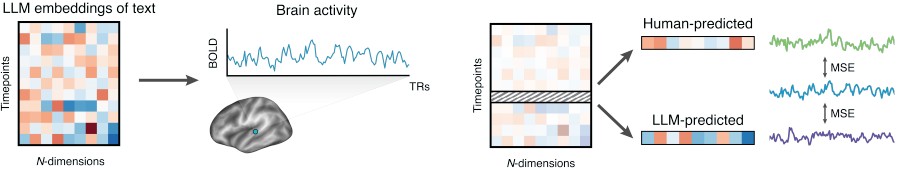

Figure 1: (a) Human participants (N=300 total) predicted upcoming words within a real-world story, presented as either spoken or written language (N=150 per modality). We compared human predictions and prediction distributions to large language models (LLM) predictions of the same stimulus. (b) Using an fMRI dataset of humans listening to the same story, we trained encoding models to predict human neural responses to spoken language. We then evaluated whether representations of human predictions of spoken language provided better predictions of neural activity than LLM-predicted words by comparing the mean-squared error (MSE) of encoding model predictions.

## 2 MATERIALS AND METHODS

Throughout this work, we leveraged an open-source dataset (LeBel et al., 2023) of human participants (N=8) that listened to real-world stories while undergoing functional magnetic resonance imaging (fMRI). All participants listened to the same auditory stories (range: 7:10 – 16:53 min) taken from The Moth podcast, and these stories were presented to participants across five separate scan sessions. In addition, one story (*wheretheressmoke*) was repeated within each of the five sessions for the purpose of model evaluation. We therefore used this story to evaluate differences between human and LLM next-word predictions and the alignment of these predictions to human brain responses.

## 2.1 BEHAVIORAL EXPERIMENT

While prior work has shown similarities between LLM next-word predictions and humans' next-word predictions for written text, no studies have investigated human predictions of upcoming words in an auditory stimulus. Given that spoken language carries extra-linguistic signals (i.e., prosody) used to infer speakers' intentions (Cole, 2015; Hellbernd & Sammler, 2016) and integrated within neural representations (Koskinen et al., 2020; Khanna et al., 2024), it becomes particularly important to directly compare human behavior across modalities. We therefore aimed to identify differences in how humans perform next-word prediction in spoken versus written language, and whether these behavioral differences drive divergence from language models (Fig. 1A).

### 2.1.1 PARTICIPANTS

Two groups of human participants were recruited for the study (N=300 total). The first group of participants (spoken condition, N=150) listened to the story without seeing the transcript. The second group of participants (written condition, N=150) viewed the story word-by-word without hearing the audio track. Participants in both conditions provided responses to the same words, and the sampled words were presented at the spoken rate to mitigate timing differences within the written condition.

### 2.1.2 LANGUAGE MODELS

We compared human responses to nine openly available LLM transformer models sourced from the *transformers* library (Wolf et al., 2020). These LLMs covered both causal language (GPT2, GPT2-XL, GPT-Neo-X, Bloom, Mistral, Llama2) and masked language (RoBERTa, Electra, XLM-ProphetNet) training objectives. For each model, we extracted the top-1 and top-5 predictions for each word in the stimulus provided the prior 100 words as context. Similar to previous studies (Goldstein et al., 2022b; Kumar et al., 2023), we also calculated the entropy of the prediction distribution for each word as a measure of prediction uncertainty.

### 2.1.3 STIMULUS

We selected the repeated story (*wheretheressmoke*) as the main stimulus for the behavioral experiment. This story was originally transcribed by the authors of the fMRI dataset, and words within the story were automatically aligned to the audio using The Penn Phonetics Lab Forced Aligner (P2FA; Yuan & Liberman (2008)). We adjusted this transcription in two ways to mitigate differences in presentation between the spoken and written language condition. First, we added punctuation to the transcription to ensure that the written text read similarly to how the original spoken version sounded. Second, and most importantly, we manually adjusted the word alignment times to ensure no overlap between subsequent words, removing any auditory cues that may advantage participants in the spoken language condition.

### 2.1.4 SELECTING CANDIDATE PREDICTION WORDS

We focused on comparing human predictions of upcoming words at moments when LLMs either succeeded or failed at performing the same task (next-word prediction). To this end, we imposed a few constraints on the words presented to human participants. First, we focused our experiment on content words (e.g., removing stop-words, named-entities, etc.) as these words are most often subject to the contextual dependencies within stories.

Second, we selectively sampled these content words based on 1) the average continuous accuracy of 5-shot model predictions (see Metrics section) and 2) the entropy of these predictions. Specifically, we identified an LLM representative of the population of tested LLMs — GPT2-XL — based on the similarity of continuous accuracy and entropy (evaluated at content words) across models. We then we divided words into four quadrants based on moments where GPT2-XL exhibited high/low continuous accuracy and high/low entropy.

Following this procedure, we sampled words from each quadrant, preserving the native distribution of words across quadrants while enforcing that the selected words were spaced apart by a minimum of 10 words. This final constraint ensured that that human participants were able to experience the story as naturally as possible without undue disruption (Vaidya et al., 2023). These words were then

divided into three presentation orders for human participants, resulting in sampling approximately 13% of all words from the story and limiting interruptions for each participant to less than 5% of the story.

### 2.1.5 METRICS

**Accuracy** Binary measures have been shown to provide a limited picture of LLM task performance (Schaeffer et al., 2023). We therefore calculated two measures of accuracy: 1) binary accuracy of the top-1 prediction (exact match) and 2) continuous accuracy, defined as the cosine similarity of the top-1 prediction and the ground-truth word. To avoid bias in computing continuous accuracy using the LLM that provided the prediction, we used a word-embedding model to estimate the semantic similarity of the predictions. We chose *fasttext* (Bojanowski et al., 2017) as the word model given its ability to provide vectors for out-of-vocabulary words.

**Predictability** Past work has evaluated the "predictability" of a word as the percentage of participants that correctly identified the upcoming word (Goldstein et al., 2022b; Vaidya et al., 2023). We also computed a "continuous predictability" score, which we defined as the average semantic similarity of human predictions to the ground-truth word. In line with prior work (Smith & Levy, 2013; Jacobs & McCarthy, 2020), we related both forms of predictability to LLM-assigned probabilities to the ground-truth word within log-odds space.

**Confidence** We assessed how LLMs and humans align in terms of the certainty of their predictions, regardless of prediction accuracy. We defined the "confidence" of a prediction as the probability assigned to the top-1 prediction by either humans or models. For both predictability and confidence, we computed Kendall's $\tau$ coefficient to assess monotonic correspondence between LLM and human predictions of a given word.

**Kullback-Leibler (KL) divergence** We examined whether LLM prediction distributions faithfully represent prediction distributions generated from human behavior. While each LLM's prediction distribution is computed over all words in its vocabulary, human prediction distributions are limited to the set of words predicted across all participants. To compare these distributions, at each predicted word we limited the LLM prediction distribution to the unique words predicted by human participants (in both conditions) and normalized the truncated distribution. We then calculated the KL divergence between human distributions to spoken and written language and LLM distributions.

### 2.2 MODELING NEURAL ACTIVITY TO NATURAL LANGUAGE

#### 2.2.1 NATURAL LANGUAGE STIMULI

All stories used within the fMRI study were transcribed by a single listener and and automatically aligned to the audio using The Penn Phonetics Lab Forced Aligner (P2FA; (Yuan & Liberman, 2008)). Alignment times of each word and phoneme within the story were manually adjusted by the original authors with Praat phonetic analysis software (Boersma, Paul, 2001).

#### 2.2.2 DATA PREPROCESSING

We preprocessed the fMRI data using *fmriprep* 23.1.4 (Esteban et al., 2019), aligning all participants anatomically to the MNI152 brain template. We then spatially smoothed the data using a 4mm Gaussian kernel and performed confound regression to remove variance in signal not associated with the stories. Lastly, we aligned the functional responses using hyperalignment (Haxby et al., 2020) to mitigate idiosyncrasies in functional topography across participants.

#### 2.2.3 STIMULUS REPRESENTATIONS

We modeled each story using four feature spaces: spectral, phoneme, semantic, and contextual features. The spectral model used a 128-dimension mel-frequency spectrogram, previously shown to be predictive of primary auditory cortex (Heelan et al., 2019; Boos et al., 2021). The phoneme model was 39-dimensions, where each dimension was a phoneme in American English defined by the CMU Pronouncing Dictionary represented as a one-hot-encoded vector. The semantic model represented

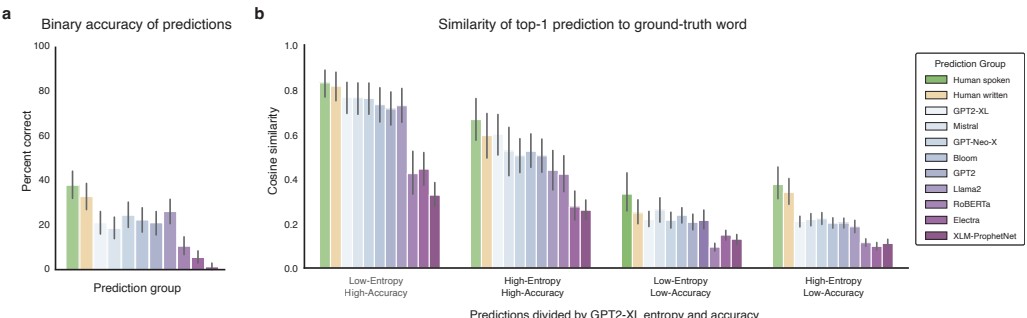

Figure 2: (a) Binary accuracy and (b) continuous accuracy (cosine similarity) of next-word predictions compared to the ground-truth word. For continuous accuracy, predictions were divided into categories based on GPT-XL continuous accuracy and entropy. In all cases, humans predicting spoken language were the most accurate, and humans in both spoken and written conditions outperformed LLMs (human vs. LLM pairwise comparisons $p < 0.001$).

each word as a 300-dimension vector within word2vec (Mikolov et al., 2013) and captured semantic relationships between words.

The contextual model comprised *N*-dimensions of a given transformer model, where *N* is the product of number of layers and the dimensionality of each layer. Importantly, this feature space captured contextual information of words beyond the purely semantic features. In line with the behavioral experiment, we used GPT2-XL as our contextual model (48 layers each of 1280 dimensions). This model has been commonly employed by other studies investigating LLM-brain relationships (Tuckute et al., 2023; Zhou et al., 2023) and shown to exhibit the strongest relationship to human representations of both spoken and written language language (Schrimpf et al., 2021).

We then adjusted this stimulus matrix to account for 1) differences in the frequency of stimulus embeddings and fMRI data and 2) variation in the hemodynamic delay across voxels (Logothetis et al., 2001; Gonzalez-Castillo et al., 2012). We first downsampled the stimulus matrix using a 3-lobe Lancosz filter to match the frequency of the fMRI data (Huth et al., 2016; LeBel et al., 2023). Then, we concatenate four copies of this matrix (spaced up to four timepoints) to the original stimulus embeddings to account for hemodynamic delays (Nishimoto et al., 2011).

### 2.2.4 TRAINING AND VALIDATING PREDICTIVE MODELS

We selected a total of 10 stories as training data, sampling two stories from each of the five fMRI sessions to avoid biasing the model to a single session. Then, for each participant, we trained voxel-wise encoding models (Huth et al., 2016) to predict a participant's neural activity from the feature-space representations of the same natural language stimulus described above. Encoding models were formalized as a banded-ridge regression (Nunez-Elizalde et al., 2019) and fit using the *himalaya* package (Dupré La Tour et al., 2022). Over the course of training, the model learned a separate regularization parameter for each feature space (including the separate transformer layers) using a leave-one-run-out cross-validation procedure (10 total folds).

To evaluate the predictive performance of the trained models, we averaged neural responses across the five separate sessions of *wheretheressmoke* and correlated the predicted and true timeseries (Huth et al., 2016). We then identified significantly predicted voxels through a block-wise permutation test. Specifically, we established a null distribution (n=1000 permutations) for each voxel by randomly shuffling the timeseries in blocks of 10 timepoints and recalculating the correlation with the predicted timeseries (LeBel et al., 2021; Jain et al., 2020).

### 2.2.5 EVALUATING DIFFERENCES BETWEEN HUMAN- AND MODEL-BASED PREDICTIONS

We compared whether human- or LLM-predicted words provided better predictions of neural responses. As the fMRI participants were presented with the stories auditorily, we focused our analysis on comparing LLM predictions to human predictions of spoken language (Fig. 1B).

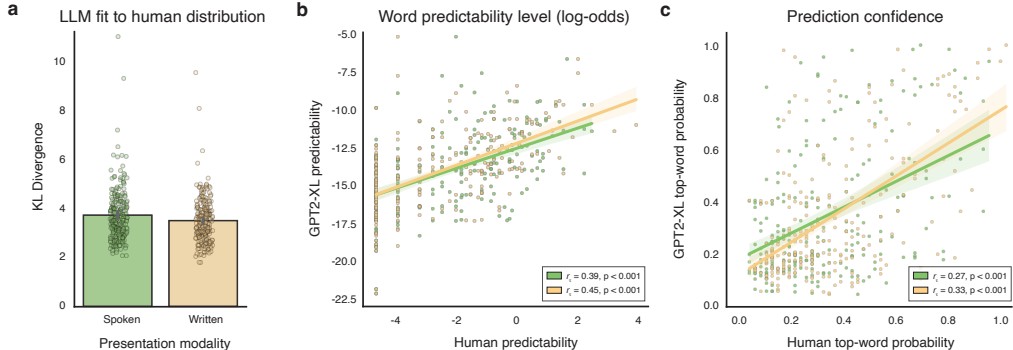

Figure 3: (a) LLM prediction distributions better fit distributions generated by humans predicting written text ($t(472) = 2.39, p = 0.017$). (b) LLM predictability levels and (c) confidence were more representative of humans predicting written text compared to humans predicting spoken language.

Similar to extracting contextual embeddings of the ground-truth stimulus, we extracted representations of each human- or LLM-predicted word given the context of the prior 100 words. We then used these embeddings to create two additional contextual feature spaces of 1) human predictions and 2) LLM predictions. Critically, we still considered the whole story, and differences in these two feature spaces are a direct result of differences in the predicted words.

We then investigated which of these two feature spaces — human- or LLM-predicted — better predicted brain activity. We specifically compared the absolute mean-squared error (MSE) predictions at the specific timepoints when a word was predicted. At each of these timepoints, we contrasted the MSE of the human- and LLM-predicted timeseries to determine which of the two representational spaces better fit brain responses.

## 3 RESULTS

### 3.1 BEHAVIORAL RESULTS

#### 3.1.1 COMPARING ACCURACY OF NEXT-WORD PREDICTIONS

We evaluated both the binary accuracy (exact match) and continuous accuracy (cosine similarity) between the top-1 prediction of humans or LLMs and the ground-truth word. Across both stimulus modalities (spoken or written), humans were more accurate than LLM predictions (Fig. 2A; all $p < 0.001$). Interestingly, even when predictions were incorrect, human predictions were more semantically similar to the ground-truth word than LLM predictions (Fig. 2B; all $p < 0.001$). Furthermore, across both forms of accuracy, humans predicting spoken language were consistently more accurate than both humans predicting written text and LLMs of spoken language consistently exhibited higher accuracy than both humans predictions of written language.

We then divided human and LLM predictions into moments (words) when GPT2-XL exhibited high versus low entropy in next-word predictions. The observed divide between human and LLM prediction accuracy was recapitulated: humans were consistently more accurate than LLMs across all accuracy-entropy quadrants of predicted words. Interestingly, this advantage for human predictions of spoken language, but not written language, was emphasized when LLMs exhibited high entropy predictions. These findings suggest that human predictions — particularly predictions of spoken language – exhibit differences in next-word prediction accuracy from LLMs, especially when LLMs are more uncertain.

#### 3.1.2 ASSESSING BEHAVIORAL ALIGNMENT OF HUMAN AND LLM PREDICTIONS

Given that humans showed higher overall accuracy (regardless of modality) than LLMs, we next aimed to understand whether the modality of presentation affected other indices of human behavioral

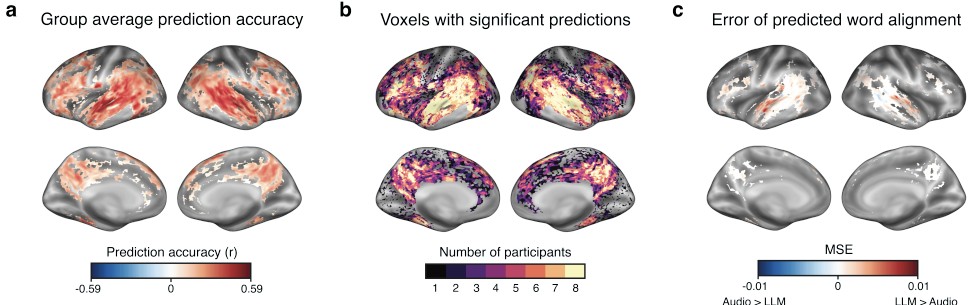

Figure 4: (a) Average performance of encoding models across participants (unthresholded). (b) Number of participants with significant encoding model predictions at each voxel ($q_{FDR} < 0.05$). (c) Human predictions of spoken language provided a better basis for predicting neural responses than LLM-predicted words as indexed by the difference in absolute mean-squared error (MSE).

alignment with LLMs. To investigate this, we selected a representative LLM (GPT2-XL) based on the fact that it had similar prediction accuracy and entropy to other tested models. We then evaluated the extent to which this LLM's prediction patterns aligned with humans' 1) prediction distributions and 2) predictability and confidence of these next-word predictions for spoken versus written language.

We first examined whether LLM prediction distributions were well-representative of the distribution of words predicted by human participants. While LLMs compute a probability distribution over all words in their vocabulary, the distribution of human predictions is limited to the words predicted across participants. To compare these distributions, we limited the LLM prediction distribution for each predicted word to the set of unique words predicted by human participants and normalized the trimmed distribution. We then calculated the Kullback-Leibler (KL) divergence between the LLM and human prediction distributions. On average, the LLM distribution exhibited significantly lower KL divergence when evaluated against the written-language distribution as compared to the spoken-language distribution (Fig. 3A; $t(472) = 2.39, p = 0.017$), indicating that model prediction patterns were more similar to humans in the written versus spoken modality.

We then compared the predictability and confidence scores assigned to the upcoming word for humans and LLMs. Across both binary and continuous predictability, humans predicting written text aligned more closely with LLM predictability scores (Fig. 3B; binary: $r_\tau = 0.45, p < 0.001$; continuous: $r_\tau = 0.41, p < 0.001$) than humans predicting spoken language (binary: $r_\tau = 0.39, p < 0.001$; continuous: $r_\tau = 0.38, p < 0.001$). The same pattern of alignment was observed for the confidence of the respective predictions. Regardless of accuracy, human showed a stronger correlation to LLM confidence in their predictions of written text (Fig. 3C; $r_\tau = 0.33, p < 0.001$) compared to predictions of spoken language ($r_\tau = 0.27, p < 0.001$). Together, these results show that the behavioral alignment between LLMs and humans deteriorates when humans process and predict spoken language.

## 3.2 PREDICTING NEURAL RESPONSES

Our final goal was to extend the behavioral findings to predictions of neural activity. To this end, we trained an ensemble of encoding models to predict human brain responses to spoken language. Given that the fMRI participants were presented with the stories as spoken language, we compared LLMs to human predictions of spoken language only.

Before comparing human- and LLM-predicted words on how well they could predict brain activity, we first validated the quality of model fits by predicting brain activity to the validation story (*wheretheressmoke*) using the ground-truth words. Across participants, LLM representations of the ground-truth words were able to predict brain activity across a large amount of cortex (Fig. 4A; unthresholded). While the voxel-wise significance of these predictions varied across participants,

trained models consistently exhibited the best predictions of brain activity within areas related to audition and language for the majority of participants (Fig. 4B; $q_{FDR} < 0.05$).

We then evaluated whether human- or LLM-predicted words provide more accurate predictions of brain activity. Accordingly, we created two separate contextual feature spaces containing either 1) human- or 2) LLM-predicted words. Specifically, we embedded each word predicted by humans or LLMs within the GPT2-XL representational space. We then used these feature spaces to generate a predicted timeseries from the trained encoding models. We compared the absolute mean-squared error (MSE) of predictions between the human- and LLM-predicted timeseries to understand which feature space (human or LLM) provides a better basis for predicting human brain activity.

We found that words from human predictions of spoken language broadly exhibited less error in predicting brain activity than words predicted by LLMs (Fig. 4C). This result provides a parallel to the divergence observed in human behavior and suggests that human predictions of spoken language are more representative of neural responses.

## 4 CONCLUSIONS

In this work, we compared human predictions of spoken and written language to predictions of large language models (LLMs). We found that humans' predictions of upcoming words diverged from LLMs' particularly when humans were asked to predict spoken language as compared with written language. Furthermore, human predictions of spoken language better explained the neural representations of participants listening to the same auditory story in the fMRI scanner. These findings are valuable in light of the many studies relating LLM surprisal and internal representations to human behavior and brain representations, suggesting both behavior and representations are less aligned than previously assumed. In particular, the behavioral and representational distinctions within the auditory modality highlight the rich, multimodal nature of spoken language, where extra-linguistic cues (i.e., prosody) may aid situational understanding to enable more accurate predictions. In future work, we plan to extend the presented methodology to other stimuli and datasets. Taken together, our results suggest that the modality by which humans perceive language alters the processing, representation, and prediction of natural language.

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
