# OpenReview forum: "Humans diverge from language models when predicting spoken language"
_ICLR.cc/2024/Workshop/Re-Align — ICLR 2024 Workshop Re-Align Poster_

### Official Review · Reviewer_EzVT · 2024-02-22
**Interesting to see a comparison of LLM next-word prediction to humans predicting both spoken and written language**

**Rating:** 2
**Fit:** 3
**Confidence:** 1

**Workshop Review:**

Summary:
- This paper compares human next-word predictions of spoken language, human next-word predictions of written language, and LLM next-word predictions of written language. The paper shows their results in terms of accuracy and semantic similarity to ground truth: humans predicting spoken language > humans predicting written language > LLM predictions.
- The paper also shows that words from human predictions of spoken language better predict the fMRI neural activity of humans listening to stories, as compared to words predicted by LLMs.
- The paper shows that LLM predictions diverge from human spoken language (their paper title) more than from human written language, suggesting that LLM behavior and representations are less aligned than previously assumed.

Strengths:
1. Tests both neural and behavioral alignment of LLMs to humans, as well as both spoken and written language
2. Tests multiple LLMs, in total 9, both causal and masked
3. Investigated multiple interesting questions related to LLM-human alignment

Weaknesses:
1. Only minor notes, see below.

Minor notes:
1. All the LLMs tested use written language, not spoken language. It would be interesting to study how speech models relate to human spoken and written language predictions.
2. The paper shows that humans predicting spoken language achieve higher accuracy than humans predicting written language (Figure 2). I was not able to tell, from reading this paper, whether this is a novel finding. (I may have missed this in the paper. If so, that is my fault).

**Reason For Not Giving Higher Score:**

N/A

**Reason For Not Giving Lower Score:**

N/A

**Reviewer Domain:**

machine learning

---

### Official Review · Reviewer_UNGp · 2024-02-24
**Good submission but the data seems somewhat incomplete**

**Rating:** 2
**Fit:** 3
**Confidence:** 2

**Workshop Review:**

This paper tries to compare the next word prediction accuracy for humans and LLMs on spoken or written language. The paper itself is fairly well written, the research question is interesting, and the methods are sound. Although this paper is taking an important initial step, it seems like there are still many unresolved issues in order to address this research question fully. It would be good if the authors address these limitations.

- The authors assessed human’s prediction accuracy for spoken and written language, and compared them to LLMs’ accuracy. However, the LLMs here seem to be all trained on written language, and it would have been more interesting if similar comparisons could be made on LLMs trained on audio.
- The authors showed that “models based on human predictions of spoken language better aligned with observed brain activity during listening compared to models based on LLM predictions”. This seems to be an unfair comparison since the fMRI data was only during listening. It would be good to see what the results would be like for brain activity during reading and how they differ from reading.
- Because of the above issues, it doesn’t seem like the current data is sufficient to support some of the authors conclusions (e.g., “spoken language carries additional information relevant to human behavior and neural representations”).
- It would be helpful for the authors to clarify some of their methods, especially how certain metrics are calculated (e.g., continuous predictability, 5-shot model predictions, entropy of predictions).

**Reason For Not Giving Higher Score:**

To fully support some of the authors' conclusions, they should include more data and results.

**Reason For Not Giving Lower Score:**

The current methods and results seems fine, and these are good initial steps towards addressing the research question.

**Reviewer Domain:**

cognitive science

---

### Official Review · Reviewer_H5W6 · 2024-02-24
**this work investigates alignment between humans and LLM in spoken language using next word prediction behavioral paradigm**

**Rating:** 3
**Fit:** 3
**Confidence:** 2

**Workshop Review:**

The work is novel provide a straightforward comparison between humans behavior and model behavior.
Clarity: the motivation, methods and results are clear.
Novelty: I think the behavioral results are novel. the fMRI results while interesting is less conclusive, and it would be useful if authors could elaborate more on what regions benefited from spoken language substitution.
Interest to the community: I believe this work is of interest to the community. In particular, using human behavior to create an upper bound on model performance is in line with the goals of the workshop

**Reason For Not Giving Higher Score:**

N/A

**Reason For Not Giving Lower Score:**

the paper is written well. the results are clear and the are not overstating. I think they could expand on neural fitting, but regardless it doesn't take away from main points of the paper

**Reviewer Domain:**

neuroscience

---

### Decision · Program_Chairs · 2024-03-02

Accept (Poster)